# Risk Factors and Surgical Management of Recurrent Herniation after Full-Endoscopic Lumbar Discectomy Using Interlaminar Approach

**DOI:** 10.3390/jcm11030748

**Published:** 2022-01-29

**Authors:** Koichiro Ono, Kazuo Ohmori, Reiko Yoneyama, Osamu Matsushige, Tokifumi Majima

**Affiliations:** 1Department of Orthopedic Surgery, Nippon Medical School, 1-1-5 Sendagi, Bunkyo-ku, Tokyo 113-8603, Japan; t-majima@nms.ac.jp; 2Center for Spinal Surgery, Nippon Koukan Hospital, 1-2-1 Koukandori, Kawasaki-ku, Kawasaki-shi 210-0852, Japan; kazuospine@gmail.com (K.O.); reikothjk@gmail.com (R.Y.); omatsushige@yahoo.co.jp (O.M.)

**Keywords:** full-endoscopic lumbar discectomy, recurrent herniation, early recurrence

## Abstract

Full-endoscopic lumbar discectomy (FED) is one of the least invasive procedures for lumbar disc herniation. Patients who receive FED for lumbar disc herniation may develop recurrent herniation at a frequency similar to conventional procedures. Reoperation and risk factors of recurrent lumbar disc herniation were investigated among 909 patients who received FED using an interlaminar approach (FED-IL). Sixty-five of the 909 patients received reoperation for recurrent herniation. Disc height, smoking, diabetes mellitus (DM), subligamentous extrusion (SE) type, and Modic change were identified as the risk factors for recurrence. Other indicators such as LL, Cobb angle, disc migration, age, sex, and body mass index (BMI) did not reach significance. Among 65 patients, reoperation was performed within 14 days following FED-IL (very early) in 7 patients, from 15 days to 3 months (early) in 14 patients, from 3 months to 1 year (midterm) in 17 patients, and after more than 1 year (late) in 27 patients. The very early group included a greater number of males, and the mean age was significantly lower in comparison to other groups. All patients in the very early group received FED-IL for reoperation. Reoperation within 2 weeks allows FED-IL to be performed without adhesion. Fusion surgery was performed on three cases in the early and midterm groups and on 10 cases in the late group, which increased over time as degenerative change and adhesion progressed. The procedure selected to treat recurrent herniation mostly depends on the surgeon’s preference. Revision FED-IL is the first choice for recurrent herniation in terms of minimizing surgical burden, whereas fusion surgery offers the advantage that discectomy can be performed through unscarred tissues. FED-IL is recommended for recurrent herniation within 2 weeks before adhesion progresses.

## 1. Introduction

Patients with lumbar disc herniation experience acute leg pain and disturbance in activities of daily living. Conservative management, in the form of medication and rest, is generally the first choice for this pathology. However, patients with intolerable pain for whom conservative treatment fails or who develop paralysis of limb muscles are considered for earlier surgical intervention. Patients who desire to return to work earlier are also candidates for discectomy. Moreover, patients who have received surgery for lumbar disc herniation show greater improvement in function and satisfaction regarding treatment than non-surgically managed patients [1], so discectomy can be recommended. Several surgical options are available for lumbar disc herniation, including open microdiscectomy [2], microendoscopic discectomy [3], and full-endoscopic lumbar discectomy (FED) [4,5,6,7]. Among these, FED is the most minimally invasive surgery as the cannula only has a diameter of 8 mm, and posterior structures are preserved [5]. The minimal trauma of FED results in a shorter operation time and hospital stay [2,8]. FED is a new technique of which the frequency and risk factors of recurrent herniation following FED remain to be elucidated. With developments in instruments and techniques, patients who have undergone FED for lumbar disc herniation may still develop recurrent herniation at a frequency similar to conventional discectomy [8]. Here, we retrospectively investigated recurrent lumbar disc herniation after FED using an interlaminar approach (FED-IL). Risk factors and surgical procedures for recurrent lumbar disc herniation that are dependent on the postoperative period are also discussed.

## 2. Materials and Methods

### 2.1. Patient Population

A total of 909 patients (630 males, 279 females) who underwent FED for lumbar disc herniation at Nippon Koukan Hospital and were followed-up for at least 1 year after primary FED-IL were retrospectively reviewed. Mean age was 49.2 years (range, 12–90 years) and mean follow-up was 46.3 months (range, 12–90.1 months). Recurrence of lumbar disc herniation was defined as relapsed disc herniation at the same disc level and on the same side as the herniation treated by the initial FED, leading to reoperation. According to these criteria, 65 patients were diagnosed with recurrent herniation (recurrent group) and the other 844 patients were registered into the non-recurrent group (Table 1). Patients with recurrent disc herniation were allocated to groups according to the interval from primary FED to reoperation: very early (VE) group, up to 14 days post-FED; early (E) group, 15 days to 3 months post-FED; midterm (M) group, 3 months to 1-year post-FED; and long-term (L) group, more than 1-year post-FED (Table 2). The mean interval from primary FED-IL to reoperation for the VE, E, M and L groups was 0.3, 1.3, 6.9, and 27.7 months, respectively (Table 2).

### 2.2. Surgical Procedures and Data Collection

#### 2.2.1. Surgical Procedure for Primary FED-IL

In all cases, primary FED-IL was performed under general anesthesia with the patient in a prone position. The entry point was marked on the symptomatic side about 1 cm lateral from the midline and caudal to the affected disc level to best utilize the interlaminar window (Figure 1a–d). After draping, a spinal needle was introduced under fluoroscopic guidance from the marked point to the caudal part of the upper lamina (Figure 1e,f). A small skin incision was made on the marked point, and the fascia was cut simultaneously (Figure 1g). A pencil dilator was inserted into the caudal part of the upper lamina (Figure 1e), and this was followed by a cannula and rigid endoscope (RIWOspine GmbH, Knittlingen, Germany). Finally, the endoscope procedure was initiated.

Soft tissues were removed using a bipolar probe (Trigger-Flex Bipolar System; Elliquence, New York, NY, USA) and forceps (Figure 2a) to expose the ligamentum flavum of the interlaminar window (Figure 2b). Drilling of the inner edge of the lamina and facet joint was performed when the interlaminar window was too narrow for cannula insertion. The ligamentum flavum was cut with an angled cutter to reach the spinal canal (Figure 2b) with additional resection to detect the lateral edge of the nerve root. The perineural membrane was removed using micro-rongeurs to precisely identify the structures (Figure 2c,d). The annulus fibrosus was opened by the dissector or bipolar probe (Figure 2e), and herniated disc material was removed piece by piece (Figure 2f). Turning the cannula prevented neural structures from causing damage during resecting herniated material (Figure 2g). After the relaxation of the nerve structure was observed (Figure 2h), the endoscope was removed, and a drainage tube was inserted through the cannula under fluoroscopic guidance. Finally, the cannula was removed.

#### 2.2.2. Re-FED-IL for Very Early Recurrent Herniation

During re-FED-IL, the endoscope was inserted using the same skin incision made for primary FED-IL. The hematoma (Figure 3a*) was removed, and relapsed herniated disc material (Figure 3b**) was detected with no adhesions. Loosening of the nerve structure marked the end of the procedure (Figure 3c***).

#### 2.2.3. Re-FED-IL for Early-to-Late Recurrent Herniation

Re-FED-IL for early-to-late recurrent herniation needs to deal with adhesion. Under endoscopic and fluoroscopic guidance, drill descending facet or lamina to reach the bottom of the spinal canal from the fresh part to avoid adhesion. Careful exposure of disc herniation was indispensable, while the adhesive nerve root and dura matter remained untouched to avoid damaging neural structures. Loosening of the structures marked the end of the procedure.

#### 2.2.4. Minimally Invasive Surgery (MIS) Posterior Lumbar Interbody Fusion (PLIF) for Recurrent Herniation

MIS-PLIF was performed as described elsewhere [9]. In brief, recurrent herniation removal and the insertion of interbody cages were performed 3–5 cm from the midline incision. Resecting the lamina or facet joint and approaching from the fresh part helped to avoid adhesion. After the midline incision closed, percutaneous pedicle screws (PPS)-rod fixation was performed. 

### 2.3. Evaluated Data 

Age, sex, body mass index (BMI), smoking, diabetes mellitus (DM), disc height, lumbar lordosis (LL), Cobb angle, migration, type of herniation (SE: subligamentous extrusion, TE: transligamentous extrusion, SQ: sequestration), and Modic change were compared between recurrent and non-recurrent groups. In addition, age, sex, BMI, reoperation procedure, 1st and 2nd operation time, and duration of postoperative hospitalization were compared among the four subgroups of recurrence. Disc height was obtained from the average of the anterior and posterior disc heights according to the Dabbs method [10]. Lumbar lordosis was measured as the angle between the superior of L1 and S1 [11].

### 2.4. Clinical Assessment

Clinical outcomes in the four subgroups were assessed preoperatively 1 month after reoperation and at the final follow-up using the Japanese Orthopedic Association (JOA) score, Oswestry Disability Index (ODI), and Visual Analog Scale (VAS) for back and leg pain. Medical records were searched for operative data and complications.

### 2.5. Ethics Approval and Consent to Participate

This study was conducted in accordance with the principles of the Declaration of Helsinki for Clinical Research [12]. The trial protocol was approved by Nippon Koukan Hospital Ethics Committee (No. 202115). Informed consent was obtained in the form of opt-out on the website.

### 2.6. Data Analysis

For numerical variables, means and standard deviations were calculated, and comparisons were made using a two-tailed Student *t*-test. Categorical variables were compared using a χ^2^ test. A *p* value of <0.05 was defined as statistically significant, and two-sided 90% confidence intervals were calculated.

## 3. Results

### 3.1. Demographics and Clinical Characteristics

Among the 909 patients who had undergone FED-IL for lumbar disc herniation, 65 patients (7.2%) experienced recurrent lumbar disc herniation and required reoperation. Mean age, BMI, and sex did not differ significantly between recurrent and non-recurrent groups, whereas smoking and DM showed higher rates in the recurrent group (Table 1).

Sixty-five patients with reoperation were allocated: 7 to the VE group, 14 to the E group, 17 to the M group, and 27 to the L group (Table 2). One patient in the L group had re-recurrent herniation. No significant differences in sex or BMI were evident among subgroups, but mean age was significantly lower in the VE group (42.9 years) in comparison to the L group (54.1 years, *p* = 0.045) (Table 2). FED-IL was performed for all 7 cases in the VE group, 9 of 14 cases in the E group, 12 of 17 cases in the M group, and 15 of 27 cases in the L group, whereas fusion surgeries were performed on 0, 3, 3, and 10 cases, respectively (Table 2). Operation time for primary FED-IL did not differ significantly between subgroups, but operation time for re-FED-IL was significantly shorter in the VE group (average 26.4 min) than in the other groups (E, 66.7 min; M, 74.9 min; L, 73.0 min; *p* < 0.01 each) (Table 2). Duration of postoperative hospitalization was similar among subgroups at primary FED-IL but was significantly shorter in the VE group (4 days) in comparison to the L group (10.7 days; *p =* 0.011) at re-FED-IL (Table 2). JOA score showed similar trends in improvement among the four subgroups after secondary FED-IL (Figure 4a). VAS scores for limb and lumbar pain were decreased at one month and final follow-up after the secondary FED-IL (Figure 4b,c). ODI was improved at the 1-month and final follow-up (Figure 4d).

### 3.2. Radiological Characteristics

Recurrent group patients had lower disc height, a greater number of subligamentous extrusion (SE) type herniation, and increased Modic change compared to the non-recurrent population. However, LL, Cobb angle, and disc migration did not reach statistical significance (Table 1).

### 3.3. Case of Very Early Group

A 51-year-old woman underwent FED-IL using a standard procedure (Figure 2) for left-sided L4/5 SE type lumbar disc herniation. Pain relief was obtained immediately after surgery, and discharge was therefore scheduled for 2 days after. However, limb pain reappeared while showering on postoperative day 2. Recurrent disc herniation was confirmed by computed tomography after myelography. Re-FED-IL was performed 7 days after the primary FED-IL (Figure 3). Operation time for this case was 12 min. Pain again improved, and she was discharged 4 days after the reoperation.

## 4. Discussion

Lumbar disc herniation is one of the common pathologies encountered in routine practice. Surgical intervention is usually indicated for unsuccessful conservative treatment. Surgery for lumbar disc herniation offers several advantages over conservative treatment, such as greater improvement of ADL [13,14] and rapid pain relief [15]. Moreover, advances in surgical techniques and equipment over recent years have enabled endoscopic lumbar discectomy, thereby reducing the burden of surgical invasiveness on the patient and facilitating a rapid return to daily life and work [16].

FED has become even more popular since FED-IL was described by Ruetten [6] and Choi [17] in 2006. Before the interlaminar approach was introduced, the transforaminal approach (FED-TF) was the only option; but this method is technically demanding for spinal surgeons who are unfamiliar with endoscopic spine surgery. Enlarged endoscopes and high-speed drills enabled an interlaminar approach that is similar to conventional open procedures. As the FED-IL is still relatively new, few long-term observational studies of large populations have been reported [4,6,18]. Ruetten et al. [6] provided the first description of full endoscopic resection of the herniated lumbar disc using an interlaminar approach on 331 patients with 2 years of follow-up. In that study, 82% of patients no longer reported leg pain after surgery, and recurrent herniation was observed in 2.7% [6]. Wasinpongwanich et al. retrospectively reviewed clinical outcomes from 545 patients for over 4 years, finding 66 recurrences (12.11%) throughout follow-up, with 7.3% of patients requiring a second operation, although the type of reoperations was not disclosed [18]. Xie reported on 479 patients with a mean follow-up of 44.3 months, with 9 cases (1.9%) showing recurrent herniation. Four cases improved with conservative treatment and five cases underwent reoperation (open surgery in three, FED-IL in two) [19]. In the present study, we followed up on 909 cases for a mean of 46.3 months, with 65 patients (7.2%) developing recurrent herniation. This was similar to previous reports with recurrence rates of 1.9–10.3% [4,19,20,21,22,23,24]. As the follow-up for patients approaches 5 years after FED-IL, the recurrence rate after FED-IL is likely to approximate the true recurrence rate.

Risk factors for recurrence after FED include old age and obesity [22,25]. Yao et al. identified age > 50 years and BMI > 25 kg/m^2^ as risk factors for recurrent herniation [25], and Kim et al. reported a mean age of 47.4 years in the recurrent group, while a mean age of 34.4 years was reported in the non-recurrent group [22]. Similarly, BMI was 24.9 kg/m^2^ in the recurrent group and 22.9 kg/m^2^ in the non-recurrent group [22]. In the present study, no difference was evident between recurrent and non-recurrent groups (Table 1). Many previous reports have described similar findings [26,27], and the apparently discrepant findings reported by Yao et al. may be explained by their use of FED-TF [22,25].

A significant correlation of smoking with the incidence of recurrent herniation has been found by several authors [28]. Nicotine plays a role in the degeneration of the intervertebral disc through narrowing the reduction in local blood flow followed by tissue hypoxia [29]. Moreover, nicotine, a small substance, can enter the intervertebral disc by diffusion and exert toxic effects directly on the intervertebral disc that causes disc degeneration [29]. In the present study, the smoking rate in the recurrent group was significantly higher in comparison to the non-recurrent group. Advanced disc degeneration by smoking may have played a role in recurrent herniation and contributed to recurrent herniation. Whether smoking cessation may reduce the recurrence of herniation remains an interesting topic for the future study.

The rate of DM patients in the recurrent group was higher in the present study than in the non-recurrent group (Table 1). Mobbs et al. also reported the rate of reoperation in the diabetic population was 28%, whereas the control group was 3.5% [30]. Interestingly, Robinson et al. reported fewer proteoglycans in the intervertebral discs of diabetic patients compared to nondiabetics, suggesting a potential mechanism underlying the higher rates in DM patients [31]. This pathological change might contribute to a decrease in the strength of the disc collagen matrix and cause susceptibility for the recurrent herniation in DM patients. Strict control of blood sugar after FED-IL is expected to reduce the recurrence of herniation.

Radiological measurements in the present study have elucidated several risk factors for recurrent herniation. Among these, disc height was lower in the recurrent group (Table 1). Kim et al. also found decreased disc height as a risk factor and hypothesized that an impaired healing process of the annulus contributes to the pathogenesis of recurrence [32]. Moreover, LL and Cobb angles were recorded, but no difference was found between groups. Large LL was suggested as the risk factor for recurrence in L5/S1 level herniation [33], whereas other levels were included in the present study. Chaojie et al. speculated wide lumbar lordosis increased shear force on the L5-S1 segment [33].

In an MRI study, subligamentous extrusion type herniation was found as the risk factor for recurrence (Table 1). Carragee et al. reported extrusion and sequestration were the end-stage in the process of fragmentation of disc material [34]. Protrusion or subligamentous extrusion-type herniation may have residual disc material that could serve as a candidate for the next herniation. Migration of disc material had no influence on recurrence, whereas Modic change was found as the risk factor for the recurrence (Table 1). Lu et al. showed recurrent herniation preferentially occurred when Modic change exists [35]. Thus, the weak connection between the cartilage and the vertebral body was the possible cause for the relapse.

Among the 65 cases, 19 patients (29.2%; VE+E groups) received reoperation within 3 months, and 46 cases (70.8%; M+L group) underwent reoperation after 3 months (Table 2). This result conflicts with the report by Kim et al. that 1001 of 2578 reoperations (38.8%) were performed within 3 months [36]. Cheng et al. followed FED-TF cases and found that 76.5% of reoperations were performed within 0.5 years after primary surgery [37]. We encountered more reoperations performed later than those reports [36,37]. This may represent differences in the approaches applied, as FED-IL or FED-TF, or in the indications for reoperation for recurrent lumbar disc herniation. Further research is needed to explain this discrepancy.

Among the four subgroups of recurrence (VE, E, M, and L groups), age, sex, and BMI were similar, except the VE group was significantly younger than the L group (Table 2). High levels of activity might cause early prolapse of disc material after the surgery. All patients in the VE group underwent FED-IL for reoperation, whereas more fusion surgeries were performed for cases with later recurrence. Surgical options for recurrent lumbar disc herniation include re-discectomy using conventional or minimally invasive techniques, with or without fusion. To determine the technique to be applied for reoperation, factors such as degeneration (including instability or deformity) and presenting symptoms (including limb or lumbar pain) are considered. Re-discectomy via a conventional [38] or minimally invasive technique [39,40] alone is the first choice for reoperation in terms of minimizing operation time, intraoperative blood loss, and total cost of the procedure [41].

Patients with back pain show significantly better improvement with fusion compared to discectomy alone, so patients with significant back pain should consider spinal fusion [41,42,43]. In addition, fusion surgeries, such as transforaminal lumbar interbody fusion, offer the advantage that discectomy can be performed through unscarred virgin tissues with no adhesions and through the lateral aspect of the dura mater with minimal retraction of the dura. While reoperation for recurrent lumbar disc herniation in previous studies [38,39,40,41,42,43] has demonstrated favorable clinical outcomes, high-quality evidence supporting indications for operative techniques remains limited, and the methods selected to treat recurrent lumbar disc herniation, for the most part, remain dependent on the preference of the surgeon.

The reoperation time for FED-IL was significantly shorter in the VE group than in the other groups. The process of scar formation and repair has been classified into three phases [44]. The first phase is the local inflammatory reaction in the first 3–5 days after surgery, involving hemostasis and coagulation. The second phase lasts 2–3 weeks for the formation of granulation tissue. The third phase involves the subsequent tissue reconstruction, transforming to scar tissue in the region of the defect [44]. The operation time for FED-IL was thus significantly shorter for the VE group than for even the E group. Reoperation performed later than 2 weeks after primary FED-IL is technically demanding for deal with adhesive soft tissues, whereas re-herniation before 2 weeks is easily removed by FED-IL without any adhesions present. In addition, the duration of hospitalization after reoperation was shorter in the VE group than in the other groups (Table 2), and clinical outcomes after reoperation were favorable (Figure 4c). These results indicate that reoperation can be positively recommended for patients with re-herniation before 2 weeks, especially since the symptoms of re-herniation are often more severe than the initial symptoms.

## 5. Study Limitations

This was a retrospective study, so key limitations in this study were selection bias and incomplete data. In the future, a prospective investigation would provide sufficient data to better clarify issues identified in this study.

## 6. Conclusions

Disc height, smoking, diabetes mellitus (DM), subligamentous extrusion (SE) type, and Modic change were identified as risk factors for recurrent lumbar disc herniation after FED-IL. The procedure selected to treat recurrent lumbar disc herniation mostly depends on the surgeon’s preference. Revision FED-IL is the first choice for the recurrent herniation in terms of minimizing surgical burden, whereas fusion surgery offers the advantage that discectomy can be performed through unscarred tissues. FED-IL is recommended for recurrent herniation within 2 weeks before adhesion progress.

## Figures and Tables

**Figure 1 jcm-11-00748-f001:**
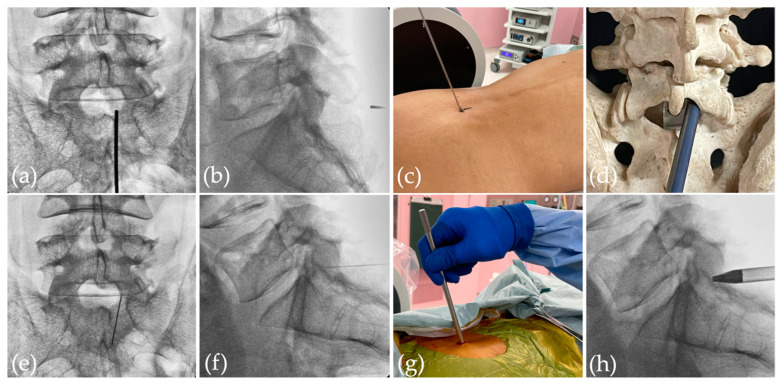
Entry point was indicated by the metal rod: (**a**) frontal view; (**b**) lateral view; (**c**) direction of endoscope on the skin; (**d**) through the interlaminar window. After draping, a spinal needle was introduced under fluoroscopic guidance from the marked point to the caudal part of the upper lamina: (**e**) anteroposterior view; (**f**) lateral view; (**g**) picture of the operative field. (**h**) Pencil dilator was inserted and placed into the interlaminar window.

**Figure 2 jcm-11-00748-f002:**
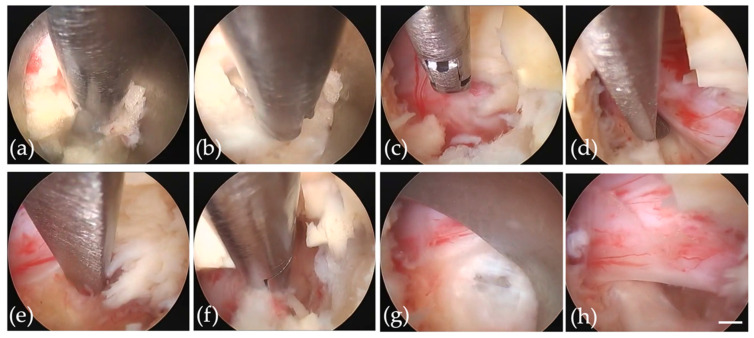
(**a**) Removal of soft tissues using Trigger-Flex Bipolar and forceps to expose the ligamentum flavum of the interlaminar window. (**b**) The ligamentum flavum was cut with an angled cutter to reach the spinal canal. (**c**) Perineural membrane was removed using micro-rongeurs. (**d**) Identification of the lateral edge of the nerve using a dissector. (**e**) The annulus was opened using a dissector or bipolar probe. (**f**) Resection of herniated disc material. (**g**) The cannula was turned to search for residual herniation. (**h**) Loosening of nerve structure was confirmed. Scale bar of 1 mm is shown as white bar.

**Figure 3 jcm-11-00748-f003:**
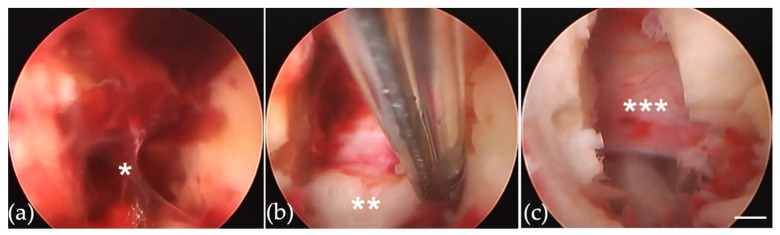
Endoscopic views of Re-FED-IL. (**a**) Hematoma (*) around the ligamentum flavum. (**b**) Relapsed herniated disc material (**) appeared from the lateral edge of the nerve root. There was no adhesion around herniation. (**c**) Loosening of nerve structure was confirmed. FED-IL = full-endoscopic discectomy, interlaminar approach. Scale bar of 1 mm is shown as white bar.

**Figure 4 jcm-11-00748-f004:**
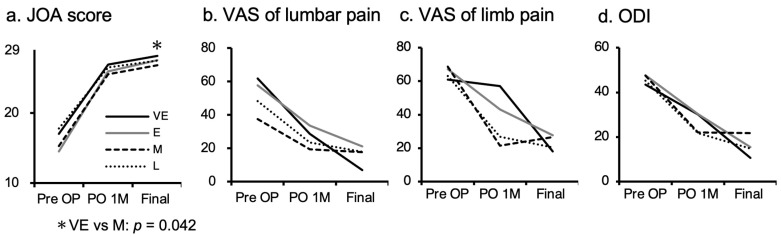
(**a**) JOA score improved similarly among the four groups after secondary FED-IL. Conversely, the VE group had significantly higher JOA scores compared to M groups at the final follow-up (* *p* = 0.042). (**b**) VAS of lumbar pain decreased similarly among the groups. (**c**) The VAS of limb pain also decreased similarly. (**d**) ODI improved equally among the groups. JOA = Japanese Orthopedic Association; ODI = Oswestry Disability Index; VAS = visual analog scale.

**Table 1 jcm-11-00748-t001:** Comparison between the recurrent and non-recurrent groups. Case number, disc height, lumbar lordosis, PI-LL, Cobb angle, migration, type of herniation (SE: subligamentous extrusion, TE: transligamentous extrusion, SQ: sequestration), Modic change, mean age, sex, mean BMI, smoking, and diabetes mellitus (DM) are shown. The recurrent group had lower disc height, higher rate of Modic change, smoking, and DM compared to the non-recurrent group. Moreover, the recurrent group tended to have more SE-type herniation than the non-recurrent group (*p* = 0.076).

Groups		Recurrent	Non-Recurrent	*p* Value
Cases (total 909)		65	844	
Mean age (years)		50.3	48.4	0.42
Sex (Male:Female)		44:21	590:254	0.71
Mean BMI (kg/m^2^)		24	26	0.36
Smoking	+	31	288	0.027
	−	34	556	
DM	+	13	84	0.011
	−	52	760	
Disc height		9.6	11.9	<0.01
Lumbar lordosis (LL)		32	34.5	0.16
Cobb angle		3.0	3.3	0.66
Migration	+	22	362	0.15
	−	43	482	
Type	SE	47	517	0.076
	non-SE (TE or SQ)	18	327	
Modic	+	31	112	< 0.01
	−	34	732	

**Table 2 jcm-11-00748-t002:** Groups of recurrent lumbar disc herniation. Patients are sorted into 4 groups depending on the interval from primary FED-IL (1st OP) to reoperation (2nd OP). Mean age, sex, mean BMI, reoperation procedure, mean 1st OP time, mean post 1st OP hospitalization days, mean 2nd OP time, and mean post 2nd OP hospitalization days are shown. * Mean age differs significantly between L and VE groups (*p* < 0.05). ** Mean 2nd OP time was significantly shorter compared to the other groups (*p* < 0.01). *** Mean post 2nd OP hospitalization days was significantly shorter compared to the other groups (*p* < 0.01). OP = operation, FEDIL = full-endoscopic lumbar discectomy, interlaminar approach, TF = transforaminal approach, MEL = microendoscopic lumbar discectomy, BMI = body mass index.

Groups	Very Early (VE)	Early (E)	Midterm (M)	Long-Term (L)
Cases (total 65)	7	14	17	27
Period from 1st to 2nd OP	0–14 days	15 days–3 months	3 months–1 year	>1 year
Mean period from 1st to 2nd OP	0.3 months	1.3 months	6.9 months	27.7 months
Mean age (years)	42.9	49.6	47.8	54.1 *
Sex (Male:Female)	6:1	9:5	9:8	20:7
Mean BMI (kg/m^2^)	23.9	22.9	24.1	24.6
Reoperation procedure (FEDIL/TF/MEL/Open/Fusion)	7/0/0/0/0	9/2/0/1/3	12/2/0/0/3	15/1/1/0/10
Mean 1st OP time (min)	59.1	61.7	71.3	72.4
Mean post-1st OP hospitalization days	7.4	3.3	3.1	4.3
Mean 2nd OP time	26.4 **	66.7	74.9	73.0
Mean post-2nd OP hospitalization days	4.0 ***	7.7	6.5	10.7

## Data Availability

The data used to support the funding of this study are available from the corresponding author upon request.

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
