# Peer review of "Risk Factors and Surgical Management of Recurrent Herniation after Full-Endoscopic Lumbar Discectomy Using Interlaminar Approach"

_jcm, 2022, doi:10.3390/jcm11030748_

Round 1

Reviewer 1 Report

I thank the authors for submitting this manuscript. The authors reported their study regarding the reoperation of interlaminar endoscopic lumbar discectomy(IELD). As a novel technique in minimally invasive spine surgery, the collection of robust evidence is necessary. The current study included 909 patients undergoing IELD, and I believed there was much valuable information in their retrospective series. However, some issues require clarification, and this paper does not merit publication before a major revision.

  1. As an observational study, the authors failed to analyze the risk factors of recurrence or differences between the first and second operations. However, the authors emphasized describing case illustrations. In my opinion, the paragraphs about case illustration, especially reoperation, were redundant. Readers will be confused and fail to understand the goal of this paper. I suggest the authors confirm the objective of their study and scientifically prove their perspectives.
  2. Most surgeons will be interested in factors or patient groups predisposing to recurrence or the difference between first and second operation strategies. The observational study can usually provide the answers, and there is still a lack of evidence regarding the issues. However, I could not see any scientific analysis regarding the critical issues. The authors did not present typical demographics, radiological data, and complication details. We are already aware of the recurrent rate of IELD from previous literature. I suggest the authors collect more clinical and radiological data, including patient occupation, complication profiles, disc height index, classification or grade of disc migration, lumbar lordosis or scoliosis measurements, etc. 
  3. According to the authors' report, 7.2% of patients encountered recurrence and underwent reoperation. Did the authors reoperate every patient with recurrence? Was there any patient with recurrence receiving conservative treatment? As for the patient undergoing a second operation, was there any second recurrence in these patients? 

Author Response

Dear Reviewer,

Thank you for inviting us to submit a revised draft of our manuscript. We also appreciate the time and effort you have dedicated to providing insightful feedback on ways to strengthen our paper. Thus, it is with great pleasure that we resubmit our article for further consideration. We have incorporated changes that reflect the detailed suggestions you have provided. We also hope that our edits and the responses we provide below satisfactorily address all the issues and concerns you and the reviewers have noted.

To facilitate your review of our revisions, the following is a point-by-point response to the questions and comments delivered in your letter dated 24 Dec 2021.

  1. As an observational study, the authors failed to analyze the risk factors of recurrence or differences between the first and second operations. However, the authors emphasized describing case illustrations. In my opinion, the paragraphs about case illustration, especially reoperation, were redundant. Readers will be confused and fail to understand the goal of this paper. I suggest the authors confirm the objective of their study and scientifically prove their perspectives.

We have additionally evaluated risk factors of recurrence, smoking, diabetes mellitus, disc height, cobb angle, lumbar lordosis, cobb angle, migration of disc fragment, type of herniation and Modic change. And add rewrote the manuscript including objective. Also, we have shortened the case illustrations.

  1. Most surgeons will be interested in factors or patient groups predisposing to recurrence or the difference between first and second operation strategies. The observational study can usually provide the answers, and there is still a lack of evidence regarding the issues. However, I could not see any scientific analysis regarding the critical issues. The authors did not present typical demographics, radiological data, and complication details. We are already aware of the recurrent rate of IELD from previous literature. I suggest the authors collect more clinical and radiological data, including patient occupation, complication profiles, disc height index, classification or grade of disc migration, lumbar lordosis or scoliosis measurements, etc. 

We have additionally collected radiological and clinical data and the risk factors for the recurrence were described.

According to the authors' report, 7.2% of patients encountered recurrence and underwent reoperation. Did the authors reoperate every patient with recurrence? Was there any patient with recurrence receiving conservative treatment?

We did not reoperate every patient with recurrence. Recurrence of lumbar disc herniation was defined as relapsed disc herniation at the same disc level and on the same side as the herniation treated by the initial FED, leading to reoperation (P2L53). There were number of the patients receiving conservative treatment. However, conservative recurrent patients could not be extracted for its vague condition.

As for the patient undergoing a second operation, was there any second recurrence in these patients? 

We observed one patient in late recurrent group had re-recurrent herniation, added description in P5L144.

Submission Date

08 December 2021

Date of this review

24 Dec 2021 09:46:58

Again, thank you for giving us the opportunity to strengthen our manuscript with your valuable comments and queries. We have worked hard to incorporate your feedback and hope that these revisions persuade you to accept our submission. 

Reviewer 2 Report

This is a retrospective study aiming to report the frequency and outcomes for revisional surgeries after primary interlaminar endoscopic lumbar discectomy. The strengths of this study include an important topic, a relatively large cohort for studying recurrence after endoscopic discectomy. The weaknesses of the study include the design and the format of the current manuscript. 

Title: The current title cannot describe this study well. Please provide a new title for this study.
Abstract: 
P1L18 The conclusion about favoring fusion surgery in recurrent disc herniation needs further explanation from these results. 
Introduction:
P1L37 What’s the meaning of “earlier” in this sentence? Please clarify.
P1L42 Not supported by new literature. FED has a higher recurrence rate compared to open and fusion.
Materials and Methods
P2L51 Please provide the ethics approval statement and number.
P2L66 Besides age, body mass index, and sex, there are many other risk factors that need to be evaluated for recurrence. Could the authors include these factors such as Modic changes, smoking, diabetes, and early physical activities, etc.?
P2L79. What is the level of significance of this study?
Results
P3L81 This section (3.1) may be moved to materials and methods.
P3L92 The position of the marker pen is different from the previous pictures. Please replace it with a picture with the correct position using the endoscope in this study for demonstration. Also “g” can be the picture with a dilator but not a needle.
P3L104 Could the author confirm that the turning of cannula is for searching of residual herniation? Also, is the "nerve structure relaxation" the end of the procedure or are there still some steps before removal of the cannula? The description of the management of the annular defect should be mentioned here as part of the section “Materials and Methods”.
P3L106 Second revision FED-IL surgeries and fusion surgery can also be briefly described.
P2L61, P4L131, P5L135 Table1, 2, and 3 can be merged.
P6161 this case demonstration is only for recurrence at a very early positive time which cannot reflect the whole picture for revision surgeries using FED-IL. Please consider changing the case demonstration or adding other cases. 
Discussion
P8L236 This part conflicts with the author's conclusion that favoring fusion surgery in recurrent HIVD after FED. Could the author clarify this part?
The authors can give some suggestions to prevent recurrence after FED based on evidence.
Conclusion
P8L270 This conclusion should be supported by a systematic review for describing the similar frequency with conventional surgeries, also the results do not clarify why fusion surgery is favorable in reoperation cases. 

Author Response

Thank you for inviting us to submit a revised draft of our manuscript. We also appreciate the time and effort you have dedicated to providing insightful feedback on ways to strengthen our paper. Thus, it is with great pleasure that we resubmit our article for further consideration. We have incorporated changes that reflect the detailed suggestions you have provided. We also hope that our edits and the responses we provide below satisfactorily address all the issues and concerns you and the reviewers have noted.

To facilitate your review of our revisions, the following is a point-by-point response to the questions and comments delivered in your letter dated 24 Dec 2021.

Title: The current title cannot describe this study well. Please provide a new title for this study.

We have changed the title from “Recurrence after full-endoscopic lumbar discectomy (interlaminar approach)” to “Risk factors and surgical strategy for recurrence after full-endoscopic lumbar discectomy using interlaminar approach”.

Abstract: 
P1L18 The conclusion about favoring fusion surgery in recurrent disc herniation needs further explanation from these results. 

We have added the explanation that fusion surgery is applied in progressed degenerative change and adhesion. Still this might be the preference of the surgeon, that discussed in P8L 239-248.

Introduction:
P1L37 What’s the meaning of “earlier” in this sentence? Please clarify.

Earlier than conventional technique. However, we have removed the sentence as no evidence supporting this.

P1L42 Not supported by new literature. FED has a higher recurrence rate compared to open and fusion.

Kevin Phan and his college (PMID: 28086154) have shown FED has similar recurrence rate to conventional discectomy. Anuj Gupta has reported fusion surgery had less recurrence (PMID 32875922), that we have changed ‘conventional procedure’ to ‘conventional discectomy’.

Materials and Methods
P2L51 Please provide the ethics approval statement and number.

We have added the section, 2.5. Ethics approval and consent to participate.

P2L66 Besides age, body mass index, and sex, there are many other risk factors that need to be evaluated for recurrence. Could the authors include these factors such as Modic changes, smoking, diabetes, and early physical activities, etc.?

We have additionally evaluated the other factors such as disc height, cobb angle, lumbar lordosis, smoking and diabetes.

P2L79. What is the level of significance of this study?

The level of significance of this study is 5. Added the comment in section, 2.6. data analysis.

Results
P3L81 This section (3.1) may be moved to materials and methods.

I have moved the section to 2.2.

P3L92 The position of the marker pen is different from the previous pictures. Please replace it with a picture with the correct position using the endoscope in this study for demonstration. Also “g” can be the picture with a dilator but not a needle.

The picture of the marker pen is modified to match the other pictures. And ‘g’ is change to the picture with a dilator.

P3L104 Could the author confirm that the turning of cannula is for searching of residual herniation? Also, is the "nerve structure relaxation" the end of the procedure or are there still some steps before removal of the cannula? The description of the management of the annular defect should be mentioned here as part of the section “Materials and Methods”.

The turning of cannula is not only for searching of residual herniation but also is the standard maneuver for removal of herniation. After the "nerve structure relaxation" is observed, the endoscope removed, and drainage is inserted through cannula under fluoroscopic guidance and finally cannula is removed. We have changed the sentence as described above.

P3L106 Second revision FED-IL surgeries and fusion surgery can also be briefly described.

We have moved methodological part of revision FED-IL of very early case to method and material, also added re-FED-IL for early to later recurrent herniation and MIS-PLIF for recurrent herniation.

P2L61, P4L131, P5L135 Table1, 2, and 3 can be merged.

We have merged the tables into table 1.

P6161 this case demonstration is only for recurrence at a very early positive time which cannot reflect the whole picture for revision surgeries using FED-IL. Please consider changing the case demonstration or adding other cases. 

We totally agree that a very early case does reflect the whole picture of revision surgeries. Revision surgeries using FED-IL have been described by Keng-Chang Liu in his manuscript, whereas revision surgery for a very early case has not been reported. In addition to the surgical procedure of revision FED-IL for very early case, we have added the description of revision FED-IL for early to late cases and MIS-PLIF for recurrent herniation (2.2.3, 2.2.4).

Discussion
P8L236 This part conflicts with the author's conclusion that favoring fusion surgery in recurrent HIVD after FED. Could the author clarify this part?

As reviewer mentioned, readers of the manuscript might misunderstand the conclusion. We have changed the conclusion of the abstract. P1L24-28.

The authors can give some suggestions to prevent recurrence after FED based on evidence.

Cessation of smoking and strict control of blood are expected to reduce recurrence. Added the sentence in P7L211,217

Conclusion
P8L270 This conclusion should be supported by a systematic review for describing the similar frequency with conventional surgeries, also the results do not clarify why fusion surgery is favorable in reoperation cases.

We have changed the conclusion as following, Disc height, smoking, diabetes mellitus (DM), subligamentous extrusion (SE) type, Modic change were identified as the risk factor for recurrence lumbar disc herniation after FED-IL. The procedure selected to treat recurrent lumbar disc herniation mostly depends on the surgeon’s preference. Revision FED-IL is the first choice for the recurrent herniation in terms of minimizing surgical burden, whereas fusion surgery offers the advantage that discectomy can be performed through unscarred tissues. FED-IL is recommended for recurrent herniation within 2 weeks, before adhesion progress.

Submission Date

08 December 2021

Date of this review

26 Dec 2021 03:59:37

Again, thank you for giving us the opportunity to strengthen our manuscript with your valuable comments and queries. We have worked hard to incorporate your feedback and hope that these revisions persuade you to accept our submission.